# COVID-19-Associated Paediatric Inflammatory Multisystem Syndrome (PIMS-TS) in Intensive Care: A Retrospective Cohort Trial (PIMS-TS INT)

**DOI:** 10.3390/children10020348

**Published:** 2023-02-10

**Authors:** Tereza Musilová, Jakub Jonáš, Tomáš Gombala, Jan David, Filip Fencl, Eva Klabusayová, Jozef Klučka, Milan Kratochvíl, Pavla Havránková, Adéla Vrtková, Kateřina Slabá, Jana Tučková, Lukáš Homola, Petr Štourač, Tomáš Vymazal

**Affiliations:** 1Department of Paediatric Anaesthesiology and Intensive Care Medicine, University Hospital Brno and Faculty of Medicine, Masaryk University, Kamenice 5, 625 00 Brno, Czech Republic; 2Department of Simulation Medicine, Faculty of Medicine, Masaryk University, Kamenice 5, 625 00 Brno, Czech Republic; 3Department of Anaesthesiology, Resuscitation and Intensive Care Medicine, 2nd Medical Faculty, Charles University and University Hospital Motol, 150 06 Prague, Czech Republic; 4Department of Pediatrics, Klinik Donaustadt, Langobardenstraße 122, 1020 Vienna, Austria; 5Department of Pediatrics, 2nd Medical Faculty, Charles University and University Hospital Motol, 150 06 Prague, Czech Republic; 6Department of Applied Mathematics, Faculty of Electrical Engineering and Computer Science, VSB Technical University of Ostrava (Czech Republic), 708 00 Ostrava, Czech Republic; 7Department of Paediatrics, University Hospital Brno and Faculty of Medicine, Masaryk University, Kamenice 5, 625 00 Brno, Czech Republic; 8Department of Pediatric Infectious Diseases, University Hospital Brno, Kamenice 5, 625 00 Brno, Czech Republic

**Keywords:** PIMS-TS, COVID-19, SARS-CoV-2, paediatric patient, post-COVID syndrome

## Abstract

Paediatric inflammatory multisystem syndrome temporally associated with COVID-19 (PIMS-TS) is a new disease in children and adolescents that occurs after often asymptomatic or mild COVID-19. It can be manifested by different clinical symptomatology and varying severity of disease based on multisystemic inflammation. The aim of this retrospective cohort trial was to describe the initial clinical presentation, diagnostics, therapy and clinical outcome of paediatric patients with a diagnosis of PIMS-TS admitted to one of the 3 PICUs. All paediatric patients who were admitted to the hospital with a diagnosis of paediatric inflammatory multisystem syndrome temporally associated with SARS-CoV-2 (PIMS-TS) during the study period were enrolled in the study. A total of 180 patients were analysed. The most common symptoms upon admission were fever (81.6%, *n* = 147), rash (70.6%, *n* = 127), conjunctivitis (68.9%, *n* = 124) and abdominal pain (51.1%, *n* = 92). Acute respiratory failure occurred in 21.1% of patients (*n* = 38). Vasopressor support was used in 20.6% (*n* = 37) of cases. Overall, 96.7% of patients (*n* = 174) initially tested positive for SARS-CoV-2 IgG antibodies. Almost all patients received antibiotics during in-hospital stays. No patient died during the hospital stay or after 28 days of follow-up. Initial clinical presentation and organ system involvement of PIMS-TS including laboratory manifestations and treatment were identified in this trial. Early identification of PIMS-TS manifestation is essential for early treatment and proper management of patients.

## 1. Introduction

Severe acute respiratory syndrome-related coronavirus 2 (SARS-CoV-2) has been a major worldwide medical issue since the first reported cases in December 2019 [1]. The worldwide spread of SARS-CoV-2 has led to a global pandemic situation, as declared by the World Health Organization (WHO) on 11 March 2020 [2,3]. According to the John Hopkins University world meter, in September 2022, more 606 million SARS-CoV-2 infections had been confirmed, with more than 6.5 million deaths reported [4]. The primary SARS-CoV-2 infection/disease (coronavirus virus disease 2019 (COVID-19)) presents as a respiratory disease. Depending on the viral subtype, initial viral load, host comorbidities, patient age and immune response, the clinical presentation of COVID-19 could lie between simple flu-like syndrome and severe acute respiratory distress syndrome (ARDS) with multiple organ failure (MOF). 

COVID-19 not only has health implications for the population; the socioeconomic effects of COVID-19 are also a significant problem. Social distancing and travel restrictions have caused the loss of many jobs in all economic sectors. Long-term school closures were intended to reduce the spread of the virus and to protect vulnerable individuals but also had various socioeconomic implications [5]. 

In the early pandemic stages, the paediatric population was considered low-risk, with only a mild clinical course of COVID-19. In the spring of 2020, about a month after the first wave of COVID-19, a new disease similar to Kawasaki disease and toxic shock syndrome began to appear [6,7,8]. A specific syndrome described by multisystemic inflammation with multiorgan involvement (Kawasaki-like) has been described in children. This syndrome was described initially as paediatric inflammatory multisystem syndrome temporally associated with COVID-19 (PIMS-TS) in the United Kingdom by the Royal College of Paediatrics and Child Health or as a multisystemic inflammatory syndrome in children (MIS-C) in the US and Europe by the Centers for Disease Control and Prevention and the European Centre for Disease Control and Prevention [9,10,11]. This unusual inflammatory disease in children that appeared after the onset of COVID-19 has been reported worldwide. It is manifested by different clinical symptomatology including fever, gastrointestinal symptoms, skin rash and varying severity of disease, including shock and involvement of the heart and coronary arteries [12]. An association with COVID-19 was assumed [13,14] because this new disease began to appear following a previous (often asymptomatic) SARS-CoV-2 infection or after contact with such an infection [15]. The majority of reported PIMS-TS cases occur within 4–6 weeks of COVID-19 infection [9,12] in a significant number of patients after a mild or even asymptomatic COVID-19 clinical course.

Only a minority of PIMS-TS patients has been tested positive for SARS-CoV-2 according to polymerase chain reaction (PCR) or antigen testing, and in the majority of cases, positive serology could be found (anti-SARS-CoV-2 antibodies) [3,16,17,18,19]. Specific anti-inflammatory treatment (derived from Kawasaki disease treatment guidelines) [20] has been recommended, such as corticosteroids, intravenous immunoglobulins, aspirin for standard clinical PIMS course or interleukin-1 (IL-1) receptor antagonist (anakinra) and interleukin-6 antagonist (tocilizumab) and tumour necrosis factor antagonist (infliximab) for severe and unresponsive disease [21,22] with very good clinical effect, as reflected by low PIMS-TS overall mortality (<2%) [3,23,24,25]. However, due to existence of several clinical definitions, variable clinical presentation and the rapid diagnostics and treatment changes over the past 2 years, significant differences could be expected. These features led to the present multicentric retrospective cohort trial with the aim of describing the potential differences in clinical presentation, early diagnostics, treatment and outcome of paediatric patients admitted to the intensive care unit with PIMS-TS. 

The aim of this retrospective cohort trial was to describe the initial clinical presentation, diagnostics, therapy and clinical outcome of paediatric patients admitted between May 2020 and May 2022 to one of three different hospitals from two countries, i.e., the Czech Republic and Austria.

## 2. Methods

This study was designed as a multicentric retrospective cohort trial. It was approved by the Ethics Committee for Multicentric Trials of the University Hospital Brno, Jihlavská 20, 625 00, Brno, Czech Republic (approval number: 06-080622/EK; Chairperson: PharmDR. Šárka Kozáková, Ph.D., MBA; date of approval: 8 June 2022) and registered on www.clinicaltrials.gov (ClinicalTrials.gov Identifier: NCT05388396). The study involved patients from three different hospitals in two countries: the Czech Republic (University Hospital Brno and Faculty of Medicine, Masaryk University, Brno; University Hospital Motol and 2nd Faculty of Medicine Prague) and Austria (Klinik Donaustadt, Vienna). All patients younger than 19 years of age who were admitted to the paediatric intensive care unit with a diagnosis of paediatric inflammatory multisystem syndrome temporally associated with SARS-CoV-2 (PIMS) TS during the study period between May 2020 and May 2022 were enrolled in the study. Exclusion criteria were open diagnosis, unconfirmed diagnosis or missing data. Diagnosis of PIMS-TS was defined by the CDC [26], the World Health Organization (WHO) [27] and the Royal College of Paediatrics and Child Health (RCPCH) criteria [10]. Due to the retrospective study design, it was not determined according to which patients were diagnosed. 

The aim of this retrospective cohort trial was to describe the initial clinical presentation, diagnostics, therapy and clinical outcome of paediatric patients with PIMS-TS admitted between May 2020 and May 2022 to one of the 3 hospital PICUs. Demographic, epidemiological data, initial clinical presentation, diagnostics, treatment and patient clinical outcome including cardiological functions were collected in each hospital from hospital information systems. The following data on the initial clinical presentation upon admission were collected: fever; dermatological involvement, i.e., skin rash or conjunctivitis; gastrointestinal involvement, i.e., abdominal pain, nausea and vomiting, diarrhoea, suspected appendicitis and appendectomy during hospital stay; neurological involvement, i.e., headache, convulsions or impaired consciousness according to the Glasgow Coma Scale (GCS); pleural effusion or pericardial effusion; and input values of physiological functions (blood oxygen levels, mean arterial pressure, body temperature and pulse). Baseline and follow-up data (one, two and three months after the onset of illness) about ECG, specifically corrected QT interval prolongation and ST wave changes, as well as echocardiography changes, specifically coronary artery abnormalities (dilation or aneurysm), tricuspid annular plane systolic excursion (TAPSE), left ventricular ejection fraction (LVEF) and cardiac wall motion abnormalities, were also collected. Subsequently, data were used for statistical analysis. Numerical variables are expressed as medians and interquartile ranges (lower and upper quartiles). Categorical variables are presented with absolute and relative frequencies (%). Spearman’s rank correlation coefficient with its confidence interval and test of significance was used to assess the association between selected numerical variables. The significance of between-group differences was tested with the Mann–Whitney test. The significance level was set to 0.05, and all statistical analyses were performed with R software (version 4.2.1) and with maximum available data. 

## 3. Results

A total of 215 patients were admitted to the paediatric intensive care unit with a suspected diagnosis of paediatric inflammatory multisystem syndrome temporally associated with SARS-CoV-2 (PIMS-TS) according to the CDC [26], the World Health Organization (WHO) [27] and the Royal College of Paediatrics and Child Health criteria [10]. Overall, 14 patients were not included for organisational reasons. A total of 201 patients were assessed for enrolment, and 180 patients were included in the final statistical analysis, with 21 patients were excluded due to the non-confirmation of the diagnosis of PIMS-TS. Figure 1 shows a flowchart of the process of patient enrolment. 

The median age was 9 years (IQR 6–11). A proportion of 65% of patients (*n* = 117) were male, and 35% of patients (*n* = 63) were female. The demographics of the cohort are described in Table 1.

The median length of hospital stay was 10 days (IQR: 8–12). The median length of PICU stay was 5 days (IQR: 1–9). The most common associated comorbidity was an increase in body mass index (BMI) above the 85th percentile (22.7%, *n* = 41): overweight (defined as 85th to less than the 95th percentile BMI [28]) was present in 9.4% (*n* = 17) of cases, and obesity (defined as 95th percentile or greater BMI [28]) was observed in 13.3% (*n* = 24) of patients. Another common comorbidity was positive allergy history in 16.7% of cases (*n* = 30). Prematurity was present in 2.8% (*n* = 5) of patients. One patient (0.6%) had bronchial asthma. None of the patients had diabetes mellitus or cancer, and none of the patients were immunocompromised. The most common admission diagnosis (32.2%, *n* = 58) was multisystem inflammatory syndrome associated with COVID-19, followed by mucocutaneous lymph node syndrome [Kawasaki] and other conditions related to polyarteritis nodosa (23.9%, *n* = 43). Table 2 summarizes the admission diagnoses with which the patients were admitted to the PICU.

A SARS-CoV-2 PCR test was performed upon admission in 92.8% of patients (*n* = 167), of which 29.3% of patients had a positive SARS-CoV-2 PCR test result (*n* = 49), with a negative result in 70.7% of cases (*n* = 118). A SARS-CoV-2 antigen test was performed upon admission in 34.4% of all patients (*n* = 62), of which 3.2% had a positive SARS-CoV-2 result (two cases), with 96.8% of patients (*n* = 60) testing negative. A proportion of 96.7% of all patients (*n* = 174) had an initial positive test result for SARS-CoV-2 IgG antibodies. The median antibody IgG level was 213.0 AU per millilitre (IQR 58.3–673.0). Only 1.1% of patients (*n* = 2) were vaccinated against SARS-CoV-2. The most common symptom upon admission was fever, defined as a measured temperature of 37 °C or greater (81.6%, *n* = 147) (37–38 °C: 26.1%, *n* = 47; greater than 38 °C: 55.5%, *n* = 100). The median body temperature upon admission was 38.4 °C (IQR: 37.2; 39.3). A proportion of 17.8% of patients had a normal body temperature upon admission (*n* = 32). The second most common symptom upon admission was skin manifestations (70.6%, *n* = 127), such as localized rash (32.2%, *n* = 58), generalized rash (30.1%, *n* = 54) or minimal rash (8.3%, *n* = 15), followed by conjunctivitis (68.9%, *n* = 124) and abdominal pain (51.1%, *n* = 92). A proportion of 28.3% of patients (*n* = 51) suffered from nausea and vomiting, and 21.7% of patients (*n* = 39) suffered from diarrhoea. A proportion of 10.6% of patients had (*n* = 19) suspected appendicitis, and 8.3% of patients (*n* = 15) had an appendectomy during their hospital stay. Most patients had a Glasgow Coma Scale (GCS) level of 13–15 (95%, *n* = 171) upon admission, and 2.8% of patients (*n* = 5) had a GCS level of 9–12 upon admission. Two patients (1.1%) had GCS levels of less than 8. In two cases (1.1%), the GCS level was unknown. A proportion of 26.1% of patients (*n* = 47) had headaches, and two patients (1.1%) had convulsions upon admission in Table 3. 

A proportion of 10.7% of patients (*n* = 19) had ECG abnormalities, specifically age-dependent corrected QT interval prolongation at baseline, 2.9% (*n* = 4) had ECG abnormalities at one month, 3.3% (*n* = 4) at two months and 0.8% (*n* = 1) at three months after the onset of illness. In 13.5% of cases (*n* = 24), ST wave changes were described at baseline, in 1.4% of cases (*n* = 2) at one month and in 0.8% of cases (*n* = 1) at two months. No patients had ECG performed during follow-up. ECG abnormalities are presented in Table 4.

Some type of coronary artery abnormality was described in 14.4% of cases (*n* = 26), with mild forms described in 11.1% of cases (*n* = 20) and moderate forms described in 3.3% of cases (*n* = 6). No patient had a severe form of coronary artery abnormality. A proportion of 16.1% of patients (*n* = 26) had a depressed left ventricular ejection fraction (LVEF) at baseline. LVEF normalized in all patients within the first month after the onset of illness. The median LVEF was 62% (IQR 55–67) at baseline, 67% (IQR 62–71) at one month, 68% (IQR 64–72) at two months and 69% (IQR 66–72) at three months after the onset of illness. Of the patients who had tricuspid annular plane systolic excursion (TAPSE), 60.7% (*n* = 17) had levels below the norm. In 32.3% of cases (*n* = 10), TAPSE remained below the norm within one month, within two months in 28% of cases (*n* = 7) and within three months after the onset of illness in 32.4% of cases (*n* = 11). Cardiac wall motion abnormalities were found in 27.4% of patients (*n* = 49) upon admission and in one case at one month after the onset of illness. No patients were monitored for echocardiographic parameters during follow-up. Echocardiographic findings are described in Table 5.

Patients often had elevated inflammatory markers upon admission, including C-reactive protein, procalcitonin and interleukin-6 levels. Abnormal blood cell counts upon admission included leukocytosis (44.4%, *n* = 80), lymphocytopenia (77.8%, *n* = 140) and thrombocytopenia (42.2%, *n* = 76). D-dimers were elevated in 98.7% of patients (*n* = 176). Some patients also had elevated cardiac markers, such as elevated troponin I in 56.7% of patients (*n* = 100) and elevated N-terminal pro-B-type natriuretic peptide in 82.8% of cases (*n* = 149). Detailed laboratory findings reported upon admission and the worst values during the hospital stay are shown in Table 6. Inflammatory markers (C-reactive protein and procalcitonin) compared with age are shown in Table 7.

A proportion of 21.1% of patients (*n* = 38) had acute respiratory failure and required oxygen or ventilatory support. Most patients required oxygen therapy (29.0%, *n* = 11), followed by high-flow nasal cannula therapy (26.3%, *n* = 10). Non-invasive ventilation was used in 18.4% of patients (*n* = 7), and intermittent positive pressure ventilation was used in 15.8% of patients (*n* = 6). Four patients (10.5%) required a combination of the abovementioned ventilatory support. The median length of duration of ventilatory support was 4 days (IQR 2–9). Vasopressor support was used in 20.6% of cases (*n* = 37), and the median length of duration of vasopressor support was 3 days (IQR 1–3). None of the patients needed renal replacement therapy during their hospital stay. 

Organ support compared with age is shown in Table 8.

Vasopressor support compared with C-reactive protein is shown in Table 9.

Overall, 98.9% (*n* = 178) patients received antibiotics. Immunosuppressive therapy included corticosteroids, intravenous immunoglobulin (IVIG) and biological therapy. Corticosteroids were prescribed in 94.4% of cases (*n* = 170). The median initial dose of corticosteroids was 23 milligrams per kilogram per day (IQR: 8–30). The median duration of corticosteroid administration was 21 days (IQR: 16–26). A proportion of 93.9% of patients (*n* = 168) received intravenous immunoglobulin. The median total dose of intravenous immunoglobulin was 2 grams per kilogram (IQR 2–2). Biological therapy included anakinra and tocilizumab. Anakinra was administered to 7.2% (*n* = 13) of patients. The median cumulative dose was 700 milligrams (IQR 250–1550). None of the patients received tocilizumab. A proportion of 88.9% of patients (*n* = 160) were reported to have received aspirin. The median dose of aspirin was 4 milligrams per kilogram per day (IQR: 3–5).

### Clinical Outcome

No patient died within 28 days; 28-day mortality was 0.0% (*n* = 0). Within 90 days, 82.2% (*n* = 148) of patients had survived, and 90-day mortality was unknown in 17.7% of cases (*n* = 32) because data were collected before 90 days of illness onset in these patients. Cardiology follow-up after three months showed complete recovery in 77.2% (*n* = 139) of patients, and 5.0% (*n* = 9) of patients had short-term residual cardiac dysfunction lasting up to one month. One patient had long-term residual cardiac dysfunction (tricuspid and mitral regurgitation, first-degree atrioventricular block). Cardiology follow-up after three months was not performed in 17.2% of cases (*n* = 31).

## 4. Discussion

This multicentre retrospective cohort trial describes the cases of 180 patients with PIMS-TS according to the CDC [26], the World Health Organization (WHO) [27] and the Royal College of Paediatrics and Child Health criteria [10]. Cases differed in terms of the age of the patient, clinical presentation and severity of illness.

The median age of patients at the onset of illness was 9 years (IQR 6–11), similar to other recently reported PIMS-TS studies [12,16,18,29,30]. One of the differences between PIMS-TS and Kawasaki disease is the higher median age of patients with PIMS-TS [12,30]. Age-specific differences between patients with PIMS-TS are likely due to age-dependent ACE2 angiotensin-converting enzyme 2 (ACE2) gene expression in the nasal epithelium, which acts as a receptor for SARS-CoV-2 to enter the cell [16,31]. Males were more represented in this cohort than females, as in previously reported studies [12,16,18].

The most common comorbidity in the cohort was an increase in body mass index (BMI) above the norm, in agreement with previously published studies in which patients with PIMS-TS are often reported to be overweight and obese [9,16]. 

Moreover, the fact that Kawasaki disease is the second most common admission diagnosis in the group of patients is probably due to the previously described similarity of PIMS-TS and Kawasaki disease; these two diseases may be interchangeable. The literature reports three phenotypes of PIMS-TS in hospitalized children, one of which fulfils the diagnostic criteria for Kawasaki disease [12,20].

The majority of patients had SARS-CoV-2 IgG-positive serology upon admission, which supports a strong causal association of PIMS-TS with the SARS-CoV-2 virus. The yield of this examination in the future is questionable when taking into account the worldwide spread of COVID-19 and the durability of plasmatic anti-Sars-COV-2 antibodies in plasma. Approximately one-third of the patients had concurrently ongoing infection confirmed by PCR test, with similar results also reported by Belhatjer et al. [32]; however in combination with extremely low positive Sars-CoV-2 antigen levels (*n* = 2, 3.2%), it seems that a positive Sars-CoV-2 PCR test does not automatically correspond to active infection or even viral particles capable of invasion and replication. Sars-CoV-2 PCR positivity could also last for several weeks or even months and does not need to be linked with active infection. This also corresponds to the statement of Fernandes et al. that the viral RNA load is usually highest at the beginning of infection and gradually decreases thereafter [33].

Furthermore, only two patients were vaccinated against COVID-19. It can be assumed that an increase in the vaccination of the paediatric population against COVID-19 could have led to a decrease in the incidence of PIMS-TS in children. Currently, vaccination is recommended for children aged 5 years and older, and a booster dose is recommended for high-risk groups [34,35]. 

COVID-19 has several indirect health effects on children and adolescents, and vaccination may help reduce the circulation of the virus in this population. Vaccination can contribute to the maintenance of the normal functioning of society [34], as COVID-19 has affected all levels of the education system. The intention of school closures was to prevent the spread of the virus; however, such closures have had widespread socioeconomic implications [5].

It is appropriate to consider offering vaccination to children at higher risk of hospitalization or at higher risk of a severe course of SARS-CoV-2 infection, for example, children with neurological diseases; Down syndrome; immunodeficiencies; malignancies; cardiac, respiratory and renal diseases; obesity; or diabetes [36,37]. Although the epidemiology of COVID-19 may change due to new variants of SARS-CoV-2, vaccination is the safest strategy to protect against SARS-CoV-2 infection, hospitalizations, long-term consequences of SARS-CoV-2 infection and death [38]. Another important factor in the fight against the COVID-19 pandemic is compliance with preventive measures [39].

The most common symptom upon admission was fever above 37 °C. However, the presence of persistent fever is one of the criteria for PIMS-TS according to the CDC [26], the World Health Organization (WHO) [27] and the Royal College of Paediatrics and Child Health criteria [10]. Therefore, the rest of the patients (17.8%) did not meet one of the diagnostic criteria for PIMS-TS. The absence of fever upon admission was probably due to the fact that patients with PIMS-TS are also transferred to these paediatric centres after initial care in smaller hospitals for further therapy or due to the administration of antipyretics before admission to the hospital. Therefore, these patients may have had a normal body temperature when they were admitted to the paediatric centre.

The incidence of organ system involvement in our cohort was consistent with previously published data. PIMS-TS is also becoming important in the differential diagnosis of abdominal pain, and even surgeons should not forget about PIMS-TS, which can mimic acute appendicitis in some patients. Children may present both clinical and laboratory signs of acute appendicitis, with inflammation of the bowel and mesentery often described, especially terminal ileitis, mesenteric lymphadenopathy or ascites upon ultrasound examination [40,41,42].

Laboratory markers of inflammation may appear to correlate with the severity of the disease [12]. This statement is also confirmed by the fact that in this cohort, patients who required vasopressor support had a higher median C-reactive protein than those who did not require vasopressor support. A statistically significant positive correlation was also found between age and CRP (Spearman’s rho (95% CI): 0.36 [0.22, 0.48], *p* < 0.001), i.e., in this cohort, older age was associated with a higher CRP level, but in the case of procalcitonin, this relationship was not proven. 

Nonetheless, children with PIMS-TS are at risk for cardiac involvement with ventricular systolic dysfunction [43]. The majority of patients had an elevated initial proBNP level above the norm (normal range up to 300 picograms per millilitre), and more than half of patients had an elevated level of troponin I (normal range up to 14 micrograms per litre), which may indicate myocardial injury. The mechanism of myocardial dysfunction is not exactly clear and is likely to be a combination of post-viral immunological reactions and systemic inflammatory response syndrome [15,43]. Initially, there were concerns about long-term cardiac involvement in patients with PIMS-TS, but over time, it has been found that persistent cardiac involvement after experiencing PIMS-TS appears to be rare [44,45,46]. As part of local PIMS-TS protocols, patients with PIMS-TS are subjected to long-term cardiology follow-up. It was proven that in the majority of patients (77.2%) from our cohort, the normalization of cardiac functions (ECG and echocardiography findings) occurred within three months after the onset of the disease. The frequent finding of cardiac involvement underlines the importance of performing a cardiological examination in all children with PIMS-TS [16].

In addition, an understanding of the pathogenesis of PIMS-TS is essential for the proper management of treatment. The probable immune dysregulation induced by SARS-CoV-2 infection has led to the inclusion of immunomodulatory drugs for therapy [16]. Almost all patients received intravenous immune globulin and systemic glucocorticoids in our cohort. This approach is consistent with published recommendations [47,48,49]. Biological therapy is recommended for patients with insufficient response to intravenous immune globulin and systemic glucocorticoid treatment. [49] It is possible to use IL-1 antagonists (anakinra), IL-6 receptor blockers (tocilizumab) or anti-TNF agents (infliximab) [49]. The choice of biological treatment should be based on the experience of specialists [49]. The high percentage of antibiotic use in this cohort can be explained by the similarity of PIMS-TS to toxic shock syndrome, in which the speed of antibiotic therapy is crucial for patient survival. Once bacterial infection has been ruled out, antibiotic treatment should be stopped [49]. 

No deaths were reported in our cohort, although deaths have been reported related to COVID-19 [35]. All patients were hospitalized in the paediatrics department with the possibility of monitoring and intensive care. This may have contributed to the early detection of disease progression and early initiation of intensive care. Alternatively, patients were transferred to the paediatrics centre from smaller hospitals after the initial provision of care. PIMS-TS is associated with severe morbidity such as acute respiratory failure, hemodynamic instability or myocardial dysfunction [3], as confirmed by the results of the present study, in which 21.1% of patients experienced acute respiratory failure that required oxygen therapy or ventilatory support, with 20.6% of patients showing vasopressor support due to hemodynamic instability and 21.1% of patients with elevated lactate above the normal value.

Not all patients required a PICU stay. It remains unknown whether there are also patients with PIMS-TS who are treated only in outpatient care without the need for hospital stay and whether PIMS-TS patients can be treated as outpatients. Further studies are needed to establish this fact.

In this retrospective study, we described the most common signs and symptoms of PIMS-TS in two hospitals in the Czech Republic and one hospital in Austria, including the laboratory findings with which patients with PIMS-TS were admitted to the hospital. The reported results may have a practical impact, especially for doctors in dealing with the early expression of suspicion of PIMS-TS and for the early initiation of treatment. This study also confirmed the correct setting of PIMS-TS intensive care protocols, as no patient deaths were reported in our cohort. In the future, especially during the next waves of COVID-19, it is advisable to increase awareness of the symptoms and manifestations of PIMS-TS not only in the medical community but also among parents and other caregivers so that children receive appropriate treatment as soon as possible.

This study has some limitations. Due to the retrospective study design, data were only available from the hospital information system. Data were collected retrospectively from three centres; therefore, it was not possible to standardize the examination methodology.

The recognition of some symptoms could be burdened by subjectivity; the examination of children was carried out by a large number of persons, and differences in diagnoses cannot be ruled out, for example, rash assessment and a subjective assessment of coronary artery abnormalities rather than according to Z-score. In this study, we also focused on the description of the initial clinical presentation and laboratory findings. We collected data on the worst laboratory findings, but we did not focus on the normalization of laboratory findings during hospitalization and further follow-up. We focused only on cardiologic long-term follow-up and not the involvement of other organ systems. We used three different diagnostic criteria to diagnose PIMS-TS: those of the CDC [26], the World Health Organization (WHO) [27] and the Royal College of Paediatrics and Child Health [10]. It was not determined according to which criteria the patients were diagnosed. Clinical management varied between hospitals, so we did not capture all variables, such as ECG and echocardiography parameters, in all patients. We did not focus on individual subtypes of PIMS-TS, although three subtypes of PIMS-TS are described in the literature [18].

PIMS-TS should be considered in patients with fever, organ dysfunction and multiorgan involvement. A positive test for SARS-CoV-2 antibodies, as the most sensitive and specific method to assess PIMS-TS, loses sensitivity over time due to the increasing number of infected persons in the population.

## 5. Conclusions

In this retrospective study, we identified the initial clinical presentation and organ system involvement of PIMS-TS, including laboratory manifestations and treatment in patients admitted to one of the three investigated PICUs. PIMS-TS is a potentially life-threatening disease. Early identification of PIMS-TS manifestation is essential for early treatment and proper management of patients. Long-term follow-up and further studies are needed to identify late manifestations and determine long-term prognosis after PIMS-TS.

## Figures and Tables

**Figure 1 children-10-00348-f001:**
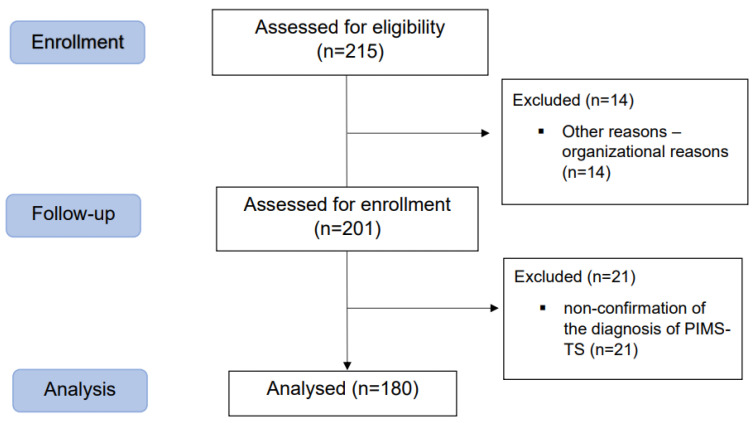
Flowchart of the process of patient enrolment.

**Table 1 children-10-00348-t001:** Cohort demographics.

	Median (IQR) or *n* (%)
Age (years)	9 (6, 11)
Height (m)	1.40 (1.19, 1.52)
Weight (kg)	31.0 (21.5, 46.0)
Sex	
Male	117 (65.0)
Female	63 (35.0)
BMI (kg/m^2^)	16.7 (15.1, 19.5)
Underweight or normal weight	138 (76.7)
Overweight	17 (9.4)
Obesity	24 (13.3)
Unknown	1 (0.6)
IQR—interquartile range	

**Table 2 children-10-00348-t002:** Admission diagnoses.

	*n* (%)
Multisystem inflammatory syndrome associated with COVID-19	58 (32.2)
Mucocutaneous lymph node syndrome [Kawasaki] and other conditions related to polyarteritis nodosa	43 (23.9)
Fever	29 (16.1)
COVID-19	15 (8.3)
Sepsis, septic shock	8 (4.4)
Gastrointestinal disease	7 (3.9)
Heart disease	5 (2.8)
Neurological disease	5 (2.8)
Lymphadenitis	3 (1.7)
Respiratory disease	2 (1.1)
Hypotension	2 (1.1)
Other	2 (1.1)
Unknown	1 (0.6)

**Table 3 children-10-00348-t003:** Summary of the input values of physiological functions.

	Median (IQR) or *n* (%)
Body temperature upon admission (°C)	38.4 (37.2, 39.3)
<37	32 (17.8)
(37, 38)	47 (26.1)
>38	100 (55.5)
Unknown	1 (0.6)
Mean arterial pressure upon admission	73 (63, 82)
Heart rate upon admission	122 (110, 140)
Oxygen saturation upon admission	98 (97, 99)
IQR—interquartile range.	

**Table 4 children-10-00348-t004:** ECG abnormalities.

	*n* (%)			
ECG	Upon Admission	1 Month	2 Months	3 Months
QTc-normal				
Yes	159 (89.3)	134 (97.1)	117 (96.7)	123 (99.2)
No	19 (10.7)	4 (2.9)	4 (3.3)	1 (0.8)
ST segment abnormalities				
Yes				
Depression	1 (0.5)	---	---	---
Elevation	6 (3.4)	1 (0.7)	1 (0.8)	---
Combination	17 (9.6)	1 (0.7)	---	---
No	154 (86.5)	136 (98.6)	120 (99.2)	124 (100.0)

**Table 5 children-10-00348-t005:** Echocardiographic findings.

	Median (IQR) or *n* (%)				
Echocardiography	Upon Admission	1 Month	2 Months	3 Months	Output
LVEF (%)	62 (55, 67)	67 (62, 71)	68 (64, 72)	69 (66, 72)	69 (65, 74)
Below the norm	26/161 (16.1)	0/134 (0.0)	0/122 (0.0)	0/122 (0.0)	0/172 (0.0)
TAPSE (cm)	1.9 (1.6, 2.1)	2.1 (1.9, 2.3)	2.1 (1.9, 2.6)	2.0 (1.9, 2.3)	---
Below the norm	17/28 (60.7)	10/31 (32.3)	7/25 (28.0)	11/34 (32.4)	---
Cardiac wall motion abnormalities					
Yes	49 (27.4)	1 (0.7)	---	---	---
No	130 (72.6)	139 (99.3)	121 (100.0)	114 (100.0)	---

**Table 6 children-10-00348-t006:** Laboratory findings.

Laboratory Findings upon Admission	*n* (%)
C-reactive protein (normal range: 0–5 mg/L)	
≤5.0	1 (0.6)
>5.0	179 (99.4)
(5.0, 200.0)	123 (68.7)
>200.0	56 (31.3)
Unknown	---
Procalcitonin (normal range: 0–0.5 ng/mL)	
≤0.5	16 (8.9)
>0.5	152 (84.4)
Unknown	12 (6.7)
Interleukin-6 (normal range: 0–7 ng/L)	
≤7	2 (1.1)
>7	59 (32.8)
Unknown	119 (66.1)
Leukocytes (normal range: 4–10 × 10 9/L)	
<4	7 (3.9)
(4, 10)	93 (51.7)
>10	80 (44.4)
Unknown	---
Lymphocytes (normal range: 1.5–3.0 × 10 9/L)	
<1.5	140 (77.8)
(1.5, 3.0)	29 (16.1)
>3.0	11 (6.1)
Unknown	---
Platelets (normal range: 150–300 × 10 9/L)	
<150	76 (42.2)
(150, 450)	100 (55.6)
>450	4 (2.2)
Unknown	---
D-dimers (normal range: 0–0.5 mg/L)	
≤0.5	2 (1.1)
>0.5	176 (97.8)
Unknown	2 (1.1)
proBNP (normal range: 0–300 pg/mL)	
≤300	25 (13.9)
>300	149 (82.8)
Unknown	6 (3.3)
Troponin I (normal range: 0–14 ng/L)	
≤14	70 (38.9)
>14	102 (56.7)
Unknown	8 (4.4)
Urea (mmol/L; age-dependant normal)	
Normal	116 (64.4)
Above the norm	63 (35.0)
Unknown	1 (0.6)
Creatinine (μmol/L; age-dependent normal)	
Normal	140 (77.8)
Above the norm	40 (22.2)
Unknown	---
Lactate (normal range: up to 2.2 mmol/L)	
≤2.2	100 (55.6)
>2.2	38 (21.1)
Unknown	42 (23.3)
	**Median (IQR)**
C-reactive protein (normal range: 0–5 mg/L)	
On admission	152.2 (94.7, 220.5)
Maximal	168.5 (114.6, 240.8)
Interleukin-6 (normal range: 0–7 ng/L)	
On admission	83.5 (30.9, 305.0)
Maximal	88.2 (35.0, 332.5)
Procalcitonin (normal range: 0–0.5 ng/mL)	
On admission	2.45 (1.20, 7.95)
Maximal	3.00 (1.27, 9.74)
Leukocytes (normal range: 4–10 × 10 9/L)	
On admission	9.80 (7.00, 12.54)
Minimal	6.00 (4.60, 7.70)
Maximal	16.45 (11.78, 21.00)
Lymphocytes (normal range: 1.5–3.0 × 10 9/L)	
On admission	0.90 (0.60, 1.43)
Platelets (normal range: 150–300 × 10 9/L)	
On admission	170.5 (121.0, 236.0)
Minimal	150.0 (102.5, 213.5)
Maximal	517.0 (406.8, 643.3)
D-dimers (normal range: 0–0.5 mg/L)	
On admission	130.1 (3.2, 1179.0)
Maximal	35.2 (4.8, 1329.0)
proBNP (normal range: 0–300 pg/mL)	
On admission	2520 (737, 6216)
Maximal	4943 (1929, 10983)
Troponin I (normal range: 0–14 ng/L)	
On admission	25.0 (6.0, 95.0)
Maximal	52.0 (14.0, 229.8)
Urea (mmol/L; age-dependant normal)	
On admission	4.50 (3.60, 9.80)
Creatinine (μmol/L; age-dependent normal)	
On admission	39.0 (24.0, 52.3)
Lactate (normal range: up to 2.2 mmol/L)	
On admission	1.6 (1.2, 2.3)
Maximal	2.3 (1.4, 3.1)

IQR—interquartile range.

**Table 7 children-10-00348-t007:** Inflammatory markers (C-reactive protein and procalcitonin) compared with age.

	rS (95%CI)	*p*
CRP	0.36 (0.22, 0.48)	<0.001
PCT	0.08 (−0.09, 0.24)	0.285
Spearman’s rank correlation coefficient, 95% confidence interval and the *p*-value of its test of significance.		

**Table 8 children-10-00348-t008:** Organ support compared with age.

	Median (IQR)	*p*
Oxygen or ventilatory support		0.145
Yes	9 (7, 12)	
No	8 (5, 11)	
Vasopressor support		<0.001
Yes	10 (9, 12)	
No	8 (5, 11)	
Median and interquartile range of age (in years) and the *p*-value of the Mann–Whitney test comparing the indicated groups of oxygen or ventilatory support and vasopressor support at the given age.		

**Table 9 children-10-00348-t009:** Vasopressor support compared with C-reactive protein.

	Median (IQR)	*p*
Vasopressor support		<0.001
Yes	227.4 (173.0, 266.4)	
No	138.6 (86.2, 183.1)	
Median and interquartile range of C-reactive protein and the *p*-value of the Mann–Whitney test comparing the indicated groups according to vasopressor support in C-reactive protein.		

## Data Availability

The data presented in this study are available upon request from the corresponding author.

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
