# Peer review of "COVID-19-Associated Paediatric Inflammatory Multisystem Syndrome (PIMS-TS) in Intensive Care: A Retrospective Cohort Trial (PIMS-TS INT)"

_children, 2023, doi:10.3390/children10020348_

Round 1

Reviewer 1 Report

First of all I want to congratulate the authors for the remarkable work. The article is well structured and has a very well conducted statistical analysis. My concerns are regarding:

- In Materials section it could be more useful to explain more details about diagnosis of PIMS-TS. (line 111)

- In Discussion section, first paragraph should belong to Introduction section, containing general data (line 287-297). The second paragraph (line 299 - 306) is a repeated part already found in Material section. The entire Discussion section should be revised as it contains extensive data already described in results section and little compared data with the existing literature. I would recommend supplementing the bibliography with at least 10-15 articles and interpret the data you have found in results.  In Discussion section there are multiple paragraphs that describe results but there is no comparison with literature data. Results seems to be repeated in this section and are not necessarily compared with other existing studies on the same topic.

Reviewer 2 Report

It was a pleasure for me to review the manuscript titled "COVID-19 associated Paediatric Inflammatory Multisystem Syndrome (PIMS-TS) in Intensive Care: retrospective cohort Trial (PIMS-TS INT)" submitted for publication in "Children" journal.

The article is pleasant to read, interesting and well written. However, there are some issues that should be fixed to merit publication in this valuable journal. Here are some comments:

1. Regarding your abstract, please try to indicate the methodology and results you got.

2. Please also a sentence about the conclusion or the main finding of your study while including recommendations

3.  Line 50, please include the full terme of the acronym "SARS-CoV-2" (and all other acronyms) in your article, as it was the first time it appeared in the text.

4. Line 62, I think you have to separate your paragraphs. In addition, before separating paragraphs, you can add a stronger background concerning the COVID-19 pandemic in the world, at least 1-2 additional sentences (eg. preventive guidlines, vaccination campagn, modeling, socio-economic impacts....etc). There is also a lack of references in your study, please consider citing these articles (and others) in the introduction of discussion section of your manuscript:

*https://doi.org/10.1136/archdischild-2021-323040

*https://doi.org/10.3390/v14122771

*https://doi.org/10.1016/j.ijsu.2020.04.018

*https://doi.org/10.1080/07853890.2022.2031274

*https://doi.org/10.3390/ijerph19159586

*https://doi.org/10.3346/jkms.2023.38.e21

5. Please include a clear objective in the last paragraph of your introduction (eg. this study aimed at.....etc)

6. Line 144, it's better to say "A total of 215 patients...etc"

7. Please refer to instructions for authors in the website of the journal to correct the layout in your table (and text too)

8. Line 287, you can also consider adding that COVID-19 have always been considered as the "eldery disease" or that older age was always associated with disease severity...etc

9. I think you need to link more the paragraphs of your discussion section, at least try to use linking words.

10. Please consider revising your manuscript by a native English or a specialist as there are some sentences that doesn't read well.

I wish the authors good luck.

Round 2

Reviewer 1 Report

good work. I have no other comments.

Reviewer 2 Report

The authors have made a great effort to revise this manuscript. They also provided detailed responses to reviewers' comment and have taken into consideration almost all my comments.

Consequently, I think the manuscript is now suitable for publication.

Good luck.